# Fruits Granola Consumption May Contribute to a Reduced Risk of Cardiovascular Disease in Patients with Stage G2–4 Chronic Kidney Disease

**DOI:** 10.3390/foods14244346

**Published:** 2025-12-17

**Authors:** Teruyuki Okuma, Hajime Nagasawa, Tomoyuki Otsuka, Hirofumi Masutomi, Satoshi Matsushita, Yusuke Suzuki, Seiji Ueda

**Affiliations:** 1Department of Nephrology, Faculty of Medicine, Juntendo University, Tokyo 113-8421, Japan; t-okuma@juntendo.ac.jp (T.O.); to-otsuka@juntendo.ac.jp (T.O.); yusuke@juntendo.ac.jp (Y.S.); se-ueda@med.shimane-u.ac.jp (S.U.); 2Division of Kidney Health and Aging, The Center for Integrated Kidney Research and Advance, Faculty of Medicine, Shimane University, Izumo 693-0021, Japan; h_masutomi@calbee.co.jp; 3Department of Life Science of Fruits Granola and Preventive Medicine, Faculty of Medicine, Juntendo University, Tokyo 113-8421, Japan; saty-m@juntendo.ac.jp; 4Department of Nephrology, Faculty of Medicine, Shimane University, Izumo 693-0021, Japan; 5Research & Development Division, Calbee, Inc., Utsunomiya 321-3231, Japan; 6Department of Cardiovascular Surgery, Faculty of Medicine, Juntendo University, Tokyo 113-8421, Japan

**Keywords:** blood pressure, chronic kidney disease (CKD), dietary fiber, fruits granola, low density lipoprotein cholesterol (LDL-C)

## Abstract

Chronic kidney disease (CKD) is estimated to affect 843.6 million people, accounting for more than 10% of the world’s population, making it a serious public health issue. Dietary therapy is important for suppressing CKD progression risk factors such as hypertension. Fruits granola (FGR), which is rich in dietary fiber, including β-glucan and polyphenols, is expected to contribute to improving the intestinal environment and providing anti-inflammatory effects. We previously reported that FGR consumption improves blood pressure and the intestinal environment in hemodialysis patients. However, the safety and efficacy of FGR for patients with moderate CKD remain unclear. Therefore, we examined the effects of FGR by replacing the breakfast of 24 patients with moderate CKD at least 5 days per week over a total of 2 months. Patients with moderate CKD who were attending outpatient appointments at the Department of Nephrology at Juntendo University Hospital and whose condition was stable were included. Patients with cancer or poor nutritional status were excluded from this study. The results revealed lower systolic blood pressure, low-density lipoprotein cholesterol (LDL-C) levels, and LDL-C/HDL-C ratios after FGR intake. Furthermore, N-acetyl-β-D-glucosaminidase (NAG), a marker of renal tubular damage, was also reduced. Regarding the intestinal environment, improved bowel movements and stool quality were observed. Based on the results of this FGR intervention study, consuming dietary fiber, which is often deficient in moderate CKD patients, may have contributed to reducing risks for cardiovascular disease and urinary tubular dysfunction through FGR intake.

## 1. Introduction

Chronic kidney disease (CKD) is a global health issue associated with rising mortality rates [1]. Currently, an estimated 843.6 million people are living with CKD, accounting for more than 10% of the world’s population and posing a serious public health problem [2,3]. Predictions indicate that CKD will likely become the fifth leading cause of life expectancy loss worldwide by 2040 [2]. Major risk factors for CKD include lifestyle-related conditions such as hypertension, diabetes, metabolic syndrome, and obesity [4,5,6,7]. In addition, recent studies have suggested that intestinal microbiota disruption may be involved in CKD progression. This is associated with increased uremic toxin production and chronic systemic inflammation, which worsens the intestinal microbiota [8]. Nutritional interventions, such as low-protein diets, dietary fiber, the Mediterranean diet, and whole-grain diets, have been shown to positively affect the composition of the intestinal microbiota and are associated with a reduced risk of CKD [8].

Cardiovascular disease (CVD) worsens life prognosis in the same manner as CKD and is a disease that frequently coexists with CKD. CVD is the leading cause of death among CKD patients, and CVD incidence increases with CKD progression [9,10]. An analysis of dietary fiber intake and its association with CVD-related and all-cause mortality in patients with metabolic syndrome revealed that higher dietary fiber intake reduced the risk of CVD-related and all-cause mortality [11]. Therefore, nutritional guidelines recommend consuming dietary fiber, but in reality, people are not reaching the target intake [12].

Fruits granola (FGR, Frugra^®^, Calbee, Inc., Tokyo, Japan) is a cereal food made primarily from oats and contains 0.5 g of salt per serving (50 g), making it a low-sodium option compared to typical Japanese or Western meals. Its consumption has been reported to improve hypertension and the intestinal environment [13]. Oats, the main ingredient in granola, contain a water-soluble dietary fiber called β-glucan, which has been shown to suppress blood glucose levels and lower low-density lipoprotein cholesterol (LDL-C) levels via meta-analysis [14,15]. We have previously reported that FGR consumption in hemodialysis patients reduces estimated salt intake, lowers blood pressure, and improves the intestinal environment, resulting in a decrease in blood indoxyl sulfate (IS) levels [16,17]. Given that dietary restrictions have been reported to result in insufficient dietary fiber intake in patients with CKD, to prevent hyperkalemia, studies examining dietary therapy for this problem are scarce, and ensuring safe dietary fiber supplementation in CKD patients has become an important issue. Therefore, we hypothesized that FGR consumption would have the potential to improve blood pressure, the intestinal environment, and lipid metabolism, which would suppress cardiovascular events in patients with moderate CKD via mechanisms similar to those observed in patients undergoing hemodialysis. Therefore, in this study, we examined the safety and efficacy of FGR consumption on blood pressure, blood glucose levels, lipid metabolism, renal tubular damage, and bowel movements for 2 months in patients with moderate CKD.

## 2. Materials and Methods

### 2.1. Subjects

Patients with moderate CKD were recruited at the Department of Nephrology, Juntendo University Hospital, for this study, and 25 patients indicated their willingness to participate in this study. Inclusion criteria were patients who were 20 years or older, had an eGFR of 15–89 mL/min, and had a stable disease status (no additional treatments or medication changes within the past six months). We excluded moderate CKD patients presenting with malignancy, active inflammation, steroid therapy, or a reduced nutritional status (Geriatric Nutritional Risk Index (GNRI) < 90). All participants provided written informed consent for participation in this study. This study was approved by the Ethics Committee of Juntendo University and was conducted according to the principles of the Declaration of Helsinki (approval number: 17-247, 22 February 2018). Furthermore, the study protocol was registered with the University Hospital Medical Information Network Clinical Trial Registry (UMIN-CRT; registration number: UMIN000031666).

### 2.2. Measurement of Clinical and Biochemical Parameters

Blood and spot urine were collected at the time of the outpatient visit. Complete blood counts were measured via flow cytometry using the XE-5000 apparatus (Sysmex, Kobe, Hyogo, Japan) [18]. Clinical chemistry tests were performed using the LABOSPECT008 Hitachi automatic analyzer (Hitachi High-Tech, Tokyo, Japan) and other instruments according to the appropriate method for each parameter, such as the enzymatic or direct method [19]. Urine tests were assayed using automated analyzers such as the DxC700AU analyzer (Beckman Coulter, Brea, CA, USA) by selecting the appropriate method for each parameter, including enzymatic assays and visible spectrophotometry [20]. IS levels were analyzed via internal-surface reversed-phase high-performance liquid chromatography (Fushimi Pharmaceutical Co., Ltd., Kagawa, Japan) [21].

Blood pressure was measured non-invasively in the brachial artery [22], and the average home blood pressure over the 7 days preceding the examination date was calculated. The patient’s nutritional status was assessed using the Geriatric Nutritional Risk Index (GNRI), which was calculated using the following formula: GNRI = [14.89 × albumin (g/dL)] + [41.7 × (body weight/ideal body weight)]. The ideal body weight of patients was calculated according to their height. For bowel health, the participants answered a questionnaire about bowel movements (Appendix A). We also checked for any changes in stool frequency or form according to the Bristol Stool Form Scale (BSS) [23].

### 2.3. Study Procedures

This is a comparison study of before and after FGR intervention. Participants’ regular breakfast was replaced with FGR (50 g) for at least 5 days per week over a total of 2 months. We conducted laboratory tests before and after the intervention (Figure 1). In this study, 50 g FGR portions per serving were packaged by Calbee Inc. and provided to subjects (Table 1), so patients easily consumed this meal without weighing it.

### 2.4. Estimated Daily Salt Intake via Tanaka’s Formula

Tanaka’s formula is a method for estimating 24 h sodium excretion (= daily salt intake) using the spot urine samples collected from patients during their visit to the Department of Nephrology at Juntendo University Hospital [24]. It is used to assess the salt intake of patients with hypertension, providing guidance for reducing salt intake. The formula is as follows: [24 h urinary Cre excretion (mg/day)] = [Weight (kg) × 14.89] + [Height (cm) × 16.14] − (Age × 2.043) − 2244.45. [24 h urinary Na excretion (mEq/day)] = 21.98 × [Spot urine Na (mEq/L) ÷ Spot urine Cre (mg/dL) ÷ 10 × 24 h urinary Cre excretion]^0.392^. [Estimated daily salt intake (g/day)] = [24 h urinary Na] ÷ 17.

### 2.5. Statistics

Statistical analyses were conducted using GraphPad Prism (version 9.5.1, GraphPad Software Inc., San Diego, CA, USA). Continuous data are presented as the mean ± standard deviation (SD), while categorical data are presented as frequencies and percentages. Within-group comparisons were conducted using the Wilcoxon signed-rank test, depending on the distribution of the data. The threshold for statistical significance was set at a *p* value of 0.05.

## 3. Results

### 3.1. Moderate CKD Patients Enrollment and the Primary Disease of CKD

A total of 25 patients with moderate CKD were recruited at the Department of Nephrology, Juntendo University Hospital (Figure 1B). There were no dropouts throughout this study. Analyses were performed on 24 subjects, and one subject was excluded due to insufficient blood collection data.

Table 2 shows the characteristics of moderate CKD patients analyzed in this study. The gender ratio was 83.3% (20 males and 4 females). The average age was 66.8 ± 9.7 years, and the average Body Mass Index (BMI) was 27.7 ± 5.1. The primary diseases due to CKD were diabetic nephropathy, nephrosclerosis, and chronic glomerulonephritis, accounting for 87.5% (*n* = 21), 8.3% (*n* = 2), and 4.2% (*n* = 1) of patients, respectively. CKD of grades 2, 3a, 3b, and 4 accounted for 4.2% (*n* = 1), 45.8% (*n* = 11), 25.0% (*n* = 6), and 25.0% (*n* = 6) of patients, respectively. There were no changes in oral medications in any of the subjects after the FGR intervention.

### 3.2. The Effect of FGR Intake on Blood Pressure

Similar to previous studies in hemodialysis patients [12,13], we investigated whether FGR intake is effective for reducing blood pressure in moderate CKD patients. Systolic blood pressure significantly decreased from 128.9 ± 11.4 at baseline to 124.3 ± 9.6 after 2 months, while diastolic blood pressure showed a tendency to decrease from 79.6 ± 7.1 at baseline to 77.4 ± 8.3 after 2 months (*p* = 0.06) (Table 3).

### 3.3. The Effects of FGR Intake on Blood Parameters

None of the 24 participants who completed the study experienced adverse events such as hyperkalemia, and no changes in hematopoietic, liver, or renal function were observed during the study period (Table 4). Regarding lipid profiles, LDL-C significantly decreased from 104.6 ± 25.4 at baseline to 98.2 ± 23.9 after 2 months; high-density lipoprotein cholesterol (HDL-C) significantly increased from 48.2 ± 13.0 at baseline to 49.4 ± 12.7 after 2 months; and Triglyceride (TG) significantly decreased from 178.3 ± 91.4 at baseline to 175.6 ± 57.4 after 2 months. The LDL/HDL ratio (LHR) significantly decreased from 2.3 ± 0.8 at baseline to 2.1 ± 0.6 after 2 months. Other blood test indices, such as electrolytes, glucose metabolism, or Hemoglobin A1c (HbA1c), showed no change. Ferritin significantly decreased from 243.4 ± 157.9 at baseline to 222.4 ± 140.5 after 2 months.

### 3.4. The Effect of FGR Intake on the Urine Test

Next, renal injury markers were examined using urine test data (Table 5). No differences were observed in urinary protein, urinary albumin, the Alb/Cre ratio, Na, K, or Cl, but urinary Cre decreased. Unexpectedly, estimated daily salt intake in moderate CKD patients increased significantly, from 8.4 ± 2.3 at baseline to 9.2 ± 1.8 after 2 months. The urinary renal tubular damage marker N-acetyl-beta-glucosaminidase (NAG) showed a significant decrease from 11.0 ± 8.9 at baseline to 7.6 ± 4.0 after 2 months.

### 3.5. The Effect of FGR Intake on Bowel Movements

To investigate the intestinal environment, we analyzed the frequency of bowel movements and stool characteristics (Table 6). The number of bowel movements per week significantly increased from 6.0 ± 2.1 at the start to 7.3 ± 2.0 after 2 months. The evaluation of stool characteristics using the BSS revealed that four patients had a disorder at the start, but all 24 patients were judged to be normal after 2 months.

## 4. Discussion

In this single-arm, before-and-after, open-label study, it was confirmed that FGR intake in patients with moderate CKD resulted in (1) a decrease in systolic blood pressure; (2) an improvement in lipid metabolism; (3) tubular damage suppression; and (4) an improvement in the bowel environment.

It has been reported that blood pressure and salt intake are correlated, with a 1.0 g reduction in salt intake resulting in a 1 mmHg reduction in blood pressure [25]. Previously, our FGR intervention studies in hemodialysis patients confirmed both a reduction in estimated salt intake and a reduction in blood pressure [16,17]. However, in this study, estimated salt intake increased after the FGR intervention (Table 5). Because spot urine samples were used in this study, it is possible that the previous meal may have affected the results. Accurately assessing salt intake requires 24 h urine collection, but this was not performed due to the significant burden on subjects. While it is possible to calculate salt intake from food records, this study did not keep food records; therefore, the subjects’ dietary habits were not known. In the future, obtaining accurate information on dietary content and urinary salt intake will be important to clarify the mechanism of blood pressure reduction.

LDL-C accumulates in the arterial wall, forming atherosclerotic plaques that increase the risk of developing CVDs [26]. In recent years, the LHR has also attracted attention. It has been reported that the LHR is positively correlated with coronary artery disease development [27]. After FGR intake, LDL-C values and the LHR decreased in our study. Soluble fiber intake reduces total cholesterol (TC) and LDL-C levels by approximately 5–10%, while changes in HDL-C or TG levels are minimal [28]. It has also been reported that increases or decreases in dietary fiber intake are deeply associated with changes in lipid profiles. [29,30]. Oats, the main ingredient in granola, contain approximately 9.4 g of dietary fiber per 100 g. Over 30% of this dietary fiber is soluble, and more than 70% of that soluble fiber is composed of β-glucan. [31]. A meta-analysis showed that consuming 3 g of β-glucan per day reduces LDL-C levels [15]. Furthermore, oats intake also has a prebiotic effect on the intestinal microbiota, suggesting the possibility of improving lipid metabolism, including TC and LDL-C, via the intestinal microbiota [32]. In a randomized, double-blind clinical trial in which oat noodles partially replaced staple foods, subjects consuming meals supplemented with oat noodles showed a significant reduction in various CVD marker levels, including the TC/HDL ratio, the LHR, and blood pressure, comparable to those in the placebo group, suggesting a potential reduction in CVD risk [33]. FGR in this study contained 0.75 g of β-glucan, suggesting that supplementation with water-soluble dietary fibers, including β-glucan, may have contributed to the reduction in LDL-C levels and the LHR.

Recently, the relationship between CKD and gut microbiota has also been reported. Alpha diversity, the bacterial count, and the diversity index have been found to be reduced in patients with CKD [34]. This study confirmed an increase in bowel movement frequency and stool-quality normalization (Table 6). CKD patients are subject to strict dietary restrictions. Therefore, dietary fiber intake has been reported to be reduced. We have previously reported that FGR may have contributed to the improvement in bowel habits and the gut environment in hemodialysis patients because FGR contained 4.5 g of dietary fiber per serving [17]. We were unable to analyze the gut microbiota in this study, and further investigation is needed. However, the fact that FGR intake reduced blood pressure despite increased salt intake may be due to an improvement in the gut environment. It is thought that the role of the gut microbiota may become one of the new strategies for preventing and managing CKD progression in the future [35].

Elevated urinary NAG levels have been reported to be associated with renal tubular damage [36] and affect the long-term prognosis of CKD [37,38]. Elevated NAG activity has also been reported in patients with hypertension [39,40], and renal dysfunction is associated with elevated urinary NAG activity and proteinuria in hypertension patients [41]. The diagnostic accuracy of NAG in predicting stage 2 or higher CKD has been shown to be good, suggesting that it may overcome the limitations of albuminuria as a marker of CKD progression risk [42]. When the kidneys are stressed, they induce ROS pathway activation, causing apoptosis of proximal tubular epithelial cells and lysosomal damage, resulting in NAG excretion in the urine after kidney injury [43]. A small-scale randomized crossover study also reported that statin administration reduced NAG levels, possibly through a reduction in LDL-C levels [44]. In this study, it is possible that lowering blood pressure and LDL-C levels could reduce kidney stress, resulting in a decrease in urinary NAG.

However, this study has several limitations: First, the study design was a single-group, before-and-after comparative study. Since FGR contains multiple ingredients, a placebo group was not possible. Also, only Japanese subjects were included, and the intervention period was short (2 months). Its applicability to ethnic groups other than the Japanese population and its contribution to long-term treatment strategies, such as whether renal prognosis improved through reduced NAG levels, are unclear. Future studies incorporating these indicators into the study design are needed to properly assess the effects of the test foods. Second, the number of subjects was small, resulting in a significant bias in gender and the primary diseases of CKD. It would be desirable to sample a larger and more diverse sample. Third, there are limitations to the method used to calculate estimated salt intake. While this study used the Tanaka formula to calculate this metric, it is susceptible to the influence of the previous day’s diet and cannot accurately represent the actual situation. For these reasons, a fourth factor, such as the implementation of a food record, should be considered. Because the participants’ dietary habits during the study were unknown, it is unclear whether their diets contributed to the reductions in blood pressure, LDL-C, the LHR, and NAG. Maintaining daily dietary records for patients with moderate CKD using meal management apps enables accurate tracking of sodium intake and other nutrient consumption, potentially leading to deeper insights. Fifth, the intestinal flora has not yet been analyzed. This may allow us to elucidate the mechanism of CKD progression, such as the relationship between the reduction in risk factors and the intestinal flora in CKD patients.

## 5. Conclusions

The results appear to show that consuming the dietary fiber contained in FGR is expected to have an effect on various factors, such as lowering blood pressure and improving lipid metabolism, tubular dysfunction, and the intestinal environment. This suggests that FGR may contribute to reducing risk factors for kidney and cardiovascular disease development and progression, not only in hemodialysis patients, as previously reported by us, but also in patients with moderate CKD, and FGR could be expected to contribute to improving life prognosis [16,17]. However, the long-term renal protective effects of FGR-mediated improvements in tubular dysfunction have not been investigated and remain a future challenge.

## Figures and Tables

**Figure 1 foods-14-04346-f001:**
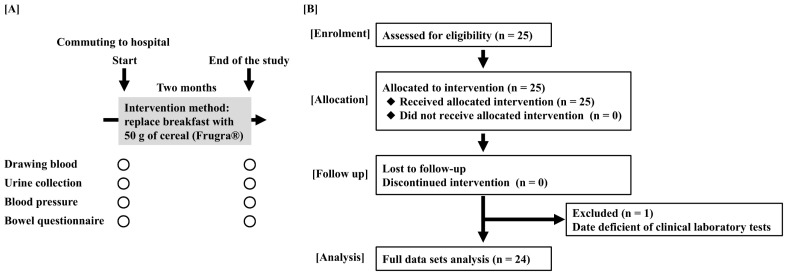
Study schedule and flow diagram of this study. (**A**) In this study, participants were recruited from patients with moderate chronic kidney disease (CKD) who consumed FGR for 2 months. (**B**) Flow diagram of this study. A total of 25 patients with CKD participated, and 24 patients were included in the analysis.

**Table 1 foods-14-04346-t001:** Nutrition information of fruits granola.

	Fruits Granola (50 g)
Energy (kcal)	220
Protein (g)	3.9
Lipid (g)	7.7
Carbohydrate glucose (g)	31.6
- Dietary fiber (g)	4.5
Salt intake (g)	0.24
Potassium (mg)	135
Calcium (mg)	16
Phosphorus (mg)	83
Iron (mg)	5.0
Vitamin A (μg)	257
Vitamin D (μg)	1.84
Vitamin B1 (mg)	0.40
Niacin (mg)	4.4
Vitamin B6 (mg)	0.44
Vitamin B12 (μg)	0.80
Folic acid (μg)	80
Pantothenic acid (mg)	1.6

**Table 2 foods-14-04346-t002:** Clinical characteristics of patients with moderate chronic kidney disease (CKD).

Patients with CKD (number)	24
Male, % (number)	83.3% (20)
Age (year, mean ± SD)	66.8 ± 9.7
Height (m, mean ± SD)	1.68 ± 0.07
Body weight (kg, mean ± SD)	78.5 ± 17.7
Body mass index (kg/m^2^, mean ± SD)	27.7 ± 5.1
Primary disease	
- Diabetic nephropathy, % (number)	87.5% (21)
- Nephrosclerosis, % (number)	8.3% (2)
- Chronic glomerulonephritis, % (number)	4.2% (1)
CKD stage	
- Grade 2, % (number)	4.2% (1)
- Grade 3a, % (number)	45.8% (11)
- Grade 3b, % (number)	25.0% (6)
- Grade 4, % (number)	25.0% (6)

SD: standard deviation.

**Table 3 foods-14-04346-t003:** Blood pressure of moderate CKD patients.

	Start	End of the Study	*p* Value
Systolic blood pressure (mmHg)	128.9 ± 11.4	124.3 ± 9.6 **	<0.01
Diastolic blood pressure (mmHg)	79.6 ± 16.4	77.4 ± 8.3	0.06

Data are shown as the mean ± standard deviation. Differences were analyzed via the Wilcoxon signed-rank test. Statistical significance was assessed relative to baseline values: ** *p* < 0.01.

**Table 4 foods-14-04346-t004:** Clinical laboratory results of moderate CKD patients.

	Start	End of the Study	*p* Value
White blood cells (count/μL)	6608 ± 2279	6467 ± 1196	0.82
Hemoglobin (g/dL)	14.2 ± 2.1	14.3 ± 2.0	0.41
Platelet (×10^4^/μL)	19.8 ± 4.5	20.5 ± 4.9	0.20
AST (U/L)	23.8 ± 13.0	22.5 ± 7.7	0.71
ALT (U/L)	25.2 ± 14.7	23.2 ± 10.7	0.26
γ-GTP (U/L)	35.6 ± 25.4	34.3 ± 23.8	0.25
Albumin (g/dL)	3.5 ± 0.3	3.7 ± 0.3	0.95
BUN (mg/dL)	24.1 ± 10.3	24.6 ± 10.7	0.68
Serum creatinine (mg/dL)	1.5 ± 0.6	1.5 ± 0.6	0.62
eGFR (mL/min/1.73m^2^)	41.4 ± 13.5	42.2 ± 14.2	0.77
Serum uric acid (mg/dL)	5.4 ± 1.4	5.3 ± 1.4	0.52
Blood glucose (mg/dL)	128.4 ± 25.7	134.2 ± 33.6	0.20
HbA1c (%)	6.5 ± 0.8	6.5 ± 0.7	0.97
LDL cholesterol (mg/dL)	104.6 ± 25.4	98.2 ± 23.9 *	0.03
HDL cholesterol (mg/dL)	48.2 ± 13.0	49.4 ± 12.7	0.15
Triglyceride (mg/dL)	178.3 ± 91.4	175.6 ± 57.4	0.69
LDL-C/HDL-C (ratio)	2.3 ± 0.8	2.1 ± 0.6 **	<0.01
Sodium (mmol/L)	140.3 ± 2.2	140.3 ± 2.5	0.96
Potassium (mmol/L)	4.4 ± 0.5	4.5 ± 0.5	0.19
Chloride (mmol/L)	104.8 ± 3.5	104.4 ± 2.8	0.40
Calcium (mg/dL)	9.4 ± 0.3	9.4 ± 0.4	0.12
Phosphorus (mg/dL)	3.3 ± 0.4	3.3 ± 0.5	0.67
Iron (μg/dL)	91.3 ± 27.5	86.4 ± 25.6	0.31
Ferritin (ng/mL)	243.4 ± 157.9	222.4 ± 140.5 *	0.02
Indoxyl sulfate (μg/mL)	2.1 ± 1.4	2.3 ± 1.7	0.43

Data are shown as the mean ± standard deviation. Differences were analyzed using the Wilcoxon signed-rank test. Statistical significance was compared against the baseline values: * *p* < 0.05 and ** *p* < 0.01. AST, aspartate aminotransferase; ALT, alanine aminotransferase; γ-GTP, γ-glutamyl transpeptidase; eGFR, estimated glomerular filtration rate; HbA1c, hemoglobin A1c; LDL-C, low-density lipoprotein cholesterol; HDL-C, high-density lipoprotein cholesterol.

**Table 5 foods-14-04346-t005:** Clinical urine tests of moderate CKD patients.

	Start	End of the Study	*p* Value
Urinary protein (mg/dL)	0.7 ± 1.0	0.7 ± 0.9	0.85
Urinary albumin (mg/gCr)	456.1 ± 694.0	408.2 ± 581.2	0.70
Urinary creatinine (mg/dL)	110.0 ± 56.4	90.1 ± 48.2 *	0.03
Alb/Cre (mg/gCr)	9.5 ± 27.5	7.2 ± 12.3	0.22
Sodium (mmol/L)	83.5 ± 46.5	83.1 ± 31.5	0.62
Potassium (mmol/L)	48.2 ± 13.0	49.4 ± 12.7	0.83
Chloride (mmol/L)	85.4 ± 45.6	89.0 ± 33.1	0.35
Estimated NaCl intake (g/day)	8.4 ± 2.3	9.2 ± 0.8 *	0.03
NAG (U/L)	11.0 ± 8.9	7.6 ± 4.0 **	<0.01

Data are shown as the mean ± standard deviation. Estimated daily salt intake was determined Via Watson’s formula. Differences were analyzed using the Wilcoxon signed-rank test. Statistical significance was compared against the baseline values: * *p* < 0.05 and ** *p* < 0.01. Alb, albumin; Cre, creatinine; NAG, N-acetyl-beta-glucosaminidase.

**Table 6 foods-14-04346-t006:** Fecal evaluation in moderate CKD patients.

	Start	End of the Study	*p* Value
Bowel movement frequency/week	6.0 ± 2.1	7.3 ± 2.0 **	<0.01
Bristol stool form scale (BSS)			
- Number of participants with disorder forms	4	0	0.04
- Number of participants with normal forms	20	24 *	

Data of bowel movement frequency are shown as the mean ± standard deviation. BSS values present the number of participants. Differences were analyzed using the Wilcoxon signed-rank test. A chi-square test was performed for participants with abnormal BSS forms and those with normal forms. Statistical significance was compared against the baseline values: * *p* < 0.05 and ** *p* < 0.01. Disorder forms: BSS 1, 2, 6, and 7. Normal forms: BSS 3–5. BSS: Bristol Stool Form Scale.

## Data Availability

The original contributions presented in this study are included in the article/Appendix A. Further inquiries can be directed to the corresponding author.

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
