# Peer review of "Fruits Granola Consumption May Contribute to a Reduced Risk of Cardiovascular Disease in Patients with Stage G2–4 Chronic Kidney Disease"

_foods, 2025, doi:10.3390/foods14244346_

Round 1
Reviewer 1 Report
Comments and Suggestions for Authors
General comment: The research article entitled “Fruits granola consumption may contribute to a reduced risk of cardiovascular disease and urinary tubular dysfunction in patients with chronic kidney disease of stage G2-4” is an interesting study. Some major corrections are required for the improvement of the manuscript.
Abstract: The Abstract presents the aim and the basic results of the study.
-Authors should better describe the type of the study, the duration, and the number of the participants
-Authors should shortly describe the inclusion and exclusion criteria
Introduction: The introduction section covers the basic theoretical background of the study.
-Authors could add a short sentence about the need to perform this study and the importance of the study.
Materials and Methods: The materials and methods are adequately presented in most cases.
-Could authors add the type of study? Eg cross-sectional observational study.
-Could authors add more details about the breakfast consumed (describe the foods)?
-Authors should refer to more data about the participants (eg age, gender), procedure of screening (eg use of questionnaire).
Results: The results of the study are analytically presented. Tables are adequate to explain the findings of the study.
Discussion: The results of study are sufficiently discussed.
Conclusion: The conclusion is adequate and summarizes the main text.
Bibliography/References: The references used by the authors cover adequately the relative scientific field and the aims of the study.
Author Response
General comment: The research article entitled “Fruits granola consumption may contribute to a reduced risk of cardiovascular disease and urinary tubular dysfunction in patients with chronic kidney disease of stage G2-4” is an interesting study. Some major corrections are required for the improvement of the manuscript.
Thank you very much for reviewing our manuscript and providing valuable comments, which have considerably helped us improve our manuscript. We revised our manuscript in accordance with your comments and suggestions. This paper focuses on the potential of Fruits granola among dietary therapy for patients with moderate CKD. We hope that you will be satisfied with the revised version of the manuscript
Abstract: The Abstract presents the aim and the basic results of the study.
-Authors should better describe the type of the study, the duration, and the number of the participants
Thank you for raising these points. The study type is comparison test of before and after FGR intervention. The Fruits granola used in this study was not a single ingredient, so a placebo food was not made. Therefore, we were unable to establish a control group. This research limitation was added in the manuscript. We have added following text.
 (page 1 , lines 27-29)
We examined the effects of FGR by replacing the breakfast of 24 patients with moderate CKD at least 5 days per week over a total of 2 months.
 (page 1, lines 36-39)
Based on the results of this FGR intervention study, consuming dietary fiber, which is often deficient in moderate CKD patients, may have contributed to reducing risks for cardiovascular disease and urinary tubular dysfunction through FGR intake.
-Authors should shortly describe the inclusion and exclusion criteria.
Thank you for your valuable comments. We have added following text in abstract and discussion.
 (page 1, lines 29-32)
Patients with moderate CKD who were attending outpatient appointments at the Department of Nephrology at Juntendo University Hospital and whose condition was stable were included. Patients with cancer or poor nutritional status were excluded from this study.
(Pages 2-3, lines 87-91)
Inclusion criteria were patients who were 20 years or older, have an eGFR of 15–89 mL/min, and have a stable disease status (no additional treatments or medication changes within the past six months). We excluded moderate CKD patients presenting with malignancy, active inflammation, steroid therapy, or a reduced nutritional status (Geriatric Nutritional Risk Index (GNRI) < 90).
Introduction: The introduction section covers the basic theoretical background of the study.
-Authors could add a short sentence about the need to perform this study and the importance of the study.
Thank you for pointing this out. We have added following text.  
(page 2 , lines 73-79)
Given that dietary restrictions have been reported to result in insufficient dietary fiber intake in patients with CKD, to prevent hyperkalemia, the studies examining dietary therapy for this problem are scarce, and ensuring safe dietary fiber supplementation in CKD patients has become an important issue. Therefore, we hypothesized that FGR consumption would have the potential to improve blood pressure, the intestinal environment, and lipid metabolism, which would suppress cardiovascular events in patients with moderate CKD via mechanisms similar to those observed in patients undergoing hemodialysis.
Materials and Methods: The materials and methods are adequately presented in most cases.
-Could authors add the type of study? Eg cross-sectional observational study.
Thank you for pointing this out. As you pointed out in the abstract section, the study type is before and after comparison test. We have added following text.
(page 3, line 117)
This is a comparison study of before and after FGR intervention.
-Could authors add more details about the breakfast consumed (describe the foods)?
Thank you for pointing out this important point. We agree that it is important to track other meals, including breakfast. We sincerely apologize, but we did not keep a food log for this study. Therefore, we have listed the importance of tracking meals as the fourth limitation in the Discussion section of the main text. We will consider this as a topic for future research.
-Authors should refer to more data about the participants (eg age, gender), procedure of screening (eg use of questionnaire).
Thank you for pointing that out. We have added a breakdown of subjects by CKD stage (Table 2). Additionally, we have added a new Supplementary File 1 regarding the content of the questionnaire used.
Results: The results of the study are analytically presented. Tables are adequate to explain the findings of the study.
Discussion: The results of study are sufficiently discussed.
Conclusion: The conclusion is adequate and summarizes the main text.
Bibliography/References: The references used by the authors cover adequately the relative scientific field and the aims of the study.
Thank you for your comments.
Reviewer 2 Report
Comments and Suggestions for Authors
The comments follow throughout the attached document.

Author Response
Comments and Suggestions for Authors
The comments follow throughout the attached document.
Thank you very much for reviewing our manuscript and providing valuable comments, which have considerably helped us improve our manuscript. We revised our manuscript in accordance with your comments and suggestions. This paper focuses on the potential of Fruits granola among dietary therapy for the patients with moderate CKD. We have replied to the comments on the file you attached. We also performed English proofreading. We hope that you will be satisfied with the revised version of the manuscript.

Round 2
Reviewer 1 Report
Comments and Suggestions for Authors
Authors have performed the suggested corrections and the ms is now improved.
Reviewer 2 Report
Comments and Suggestions for Authors
After making the indicated changes, your manuscript has been significantly improved and can be accepted for publication.